# GaitSnippet: Gait Recognition Beyond Unordered Sets and Ordered Sequences

**Saihui Hou[1], Chenye Wang[1], Wenpeng Lang[1], Zhengxiang Lan[1], & Yongzhen Huang[1,2*]**
[1]School of Artificial Intelligence, Beijing Normal University [2]WATRIX.AI
`{housaihui, huangyongzhen}@bnu.edu.cn`
`{chenye.wang, wenpenglang, zhengxianglan}@mail.bnu.edu.cn`

## Abstract

Recent advancements in gait recognition have significantly enhanced performance by treating silhouettes as either an unordered set or an ordered sequence. However, both set-based and sequence-based approaches exhibit notable limitations. Specifically, set-based methods tend to overlook short-range temporal context for individual frames, while sequence-based methods struggle to capture long-range temporal dependencies effectively. To address these challenges, we draw inspiration from human identification and propose a new perspective that conceptualizes human gait as a composition of individualized actions. Each action is represented by a series of frames, randomly selected from a continuous segment of the sequence, which we term a **snippet**. Fundamentally, the collection of snippets for a given sequence enables the incorporation of multi-scale temporal context, facilitating more comprehensive gait feature learning. Moreover, we introduce a non-trivial solution for snippet-based gait recognition, focusing on Snippet Sampling and Snippet Modeling as key components. Extensive experiments on four widely-used gait datasets validate the effectiveness of our proposed approach and, more importantly, highlight the potential of gait snippets. For instance, our method achieves the rank-1 accuracy of 77.5% on Gait3D and 81.7% on GREW using a 2D convolution-based backbone.

## 1 Introduction

Gait recognition aims to identify individuals based on their unique walking patterns. This technique can be performed at a distance without the explicit cooperation of the subjects, making it highly applicable in areas such as social security Rida et al. (2019), human-computer interaction Zhu et al. (2022), and health monitoring Bortone et al. (2021), *etc*. Silhouettes are commonly used as input, as they effectively eliminate clothing texture while remaining robust under low-resolution conditions.

In the gait recognition literature, early studies typically aggregated silhouettes into a template, such as Gait Energy Image Han & Bhanu (2005), which, although simple, inevitably sacrifices fine-grained details. Recent research predominantly treats silhouettes either as an unordered set or an ordered sequence, leveraging deep neural networks to extract gait features. Specifically, set-based methods Chao et al. (2019); Hou et al. (2020; 2021; 2022b) assume that the appearance of a silhouette inherently contains its positional information, rendering the order information unnecessary. The pioneering GaitSet Chao et al. (2019), a representative of this category, significantly improves performance over template-based methods and demonstrates resilience to frame permutations. In contrast, sequence-based methods Lin et al. (2020; 2021); Huang et al. (2021b;a) treat a sequence of silhouettes as a video, utilizing 3D Tran et al. (2015) or P3D Qiu et al. (2017) convolutions, along with their variants Lin et al. (2020), to extract both spatial and temporal features.

Despite the significant performance gains of recent advancements, both set-based and sequence-based paradigms exhibit notable limitations. First, in set-based methods, feature extraction in the backbone, typically performed using 2D convolution, processes each silhouette independently, lacking awareness of short-range temporal context between adjacent frames. Second, in sequence-based

---

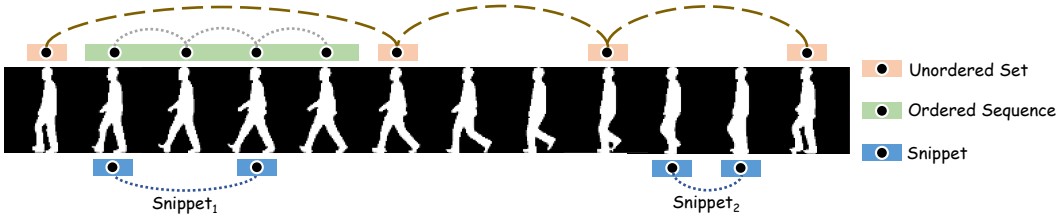

Figure 1: Illustration of gait snippets in comparison to unordered sets and ordered sequences. Best viewed in color.

methods, feature extraction primarily relies on 3D/P3D convolutions or their variants, with a limited number of continuous frames (*e.g.*, 30) sampled from each sequence during training. This approach significantly hinders the ability to model long-range temporal dependencies, especially in long sequences (*e.g.*, those with more than 200 frames in real-world benchmarks Zheng et al. (2022)). This raises a critical question: *Is there an alternative paradigm for extracting gait features from silhouettes that addresses these limitations?*

In this work, we propose a new perspective on gait recognition inspired by human cognition, arguing that identification often depends on key actions in a few adjacent frames—not a full cycle. This aligns with the biological finding that "*recognition is possible for stimuli lasting a fraction of a full walking cycle*" Giese & Poggio (2003). Motivated by this insight, we propose to conceptualize human gait as *a composition of individualized actions*. Specifically, as illustrated in Figure 1, we represent an action using several frames randomly selected from a continuous segment of the sequence, which we term a **snippet**. This approach allows an individual's walking pattern to be described as the union of snippets derived from the same sequence. Gait snippets offer two notable conceptual advantages: (1) Compared to unordered sets, snippets facilitate the incorporation of short-range temporal context for frame-level feature extraction. (2) Compared to ordered sequences, snippets enable the capture of long-range temporal dependencies within a long sequence.

Building on these insights, we focus on snippet-based gait recognition and address two critical challenges: (a) *How to sample snippets during the input phase for training and inference?* (b) *How to effectively model snippet-based inputs for gait recognition?* In this work, we propose an efficient yet effective solution, marking the first attempt to systematically tackle these challenges.

*Regarding Snippet Sampling*, given a sequence of silhouettes, we treat it as non-continuous due to imperfect upstream processing and various occlusions Fan et al. (2023b), but we assume that the relative order of frames is preserved. This order is used to divide the sequence into non-overlapping segments of equal length. For training, we randomly select a subset of frames from each segment to form a snippet representing an individualized action, with the number of snippets generally fewer than the number of segments. For inference, all frames from each segment are used to construct a snippet, and all snippets from a sequence are utilized to match the probe and gallery. *In terms of Snippet Modeling*, we design an efficient framework to address three core challenges: (1) **Intra-Snippet Modeling**: We introduce a Snippet Block where a non-parametric pooling operation captures local temporal context within a snippet, merging it with frame-level features through a residual connection. (2) **Cross-Snippet Modeling**: We treat all snippets within a sequence as an unordered set, employing Set Pooling to derive sequence-level representations based on intra-snippet modeling. (3) **Snippet-Level Supervision**: Representing gait through snippets enables hierarchical representations at both the sequence and snippet levels. In addition to sequence-level loss, we introduce snippet-level supervision to further enhance training.

In summary, the main contributions are threefold:

(1) We introduce a new perspective on gait recognition, organizing a sequence of silhouettes as a union of snippets to characterize the walking pattern.

(2) We pioneer snippet-based gait recognition, designing a comprehensive solution that includes Snippet Sampling and Snippet Modeling.

(3) Extensive experimental results demonstrate the potential of gait snippets, with our approach achieving the rank-1 accuracy of 77.5% on Gait3D Zheng et al. (2022) and 81.7% on GREW Zhu et al. (2021) using a 2D convolutional backbone.

## 2 RELATED WORK

**Gait Recognition** We address the fundamental challenges in the modeling paradigm for gait recognition by using silhouettes as input. In early studies Han & Bhanu (2005); Wang et al. (2010), silhouettes were usually aggregated into templates. More recent advancements have treated silhouettes as either unordered sets Chao et al. (2019); Hou et al. (2020; 2021; 2022b); Fan et al. (2023c) or ordered sequences Lin et al. (2020); Fan et al. (2020); Lin et al. (2021); Huang et al. (2021b;a); Ma et al. (2023); Dou et al. (2023); Wang et al. (2023a;c) for feature learning. Below, we briefly review representative methods within these two subcategories.

(1) *Unordered Sets*: GaitSet Chao et al. (2019) is the first to introduce set-based feature learning for silhouettes, using horizontal splits of feature maps to learn multiple part representations. GLN Hou et al. (2020) merges multi-stage features for set-based modeling, focusing on reducing feature dimensionality to enhance recognition performance. GaitBase Fan et al. (2023c) and its deeper variant, DeepGaitV2-2D Fan et al. (2023a), present a robust ResNet-like 2D backbone, achieving competitive performance across various benchmarks.

(2) *Ordered Sequences*: GaitGL Lin et al. (2021) utilizes 3D convolution to blend local and global feature extraction in its architecture. GaitGCI Dou et al. (2023) introduces a counterfactual intervention to mitigate the effects of confounding factors while using dynamic convolution for factual/counterfactual attention generation. DyGait Wang et al. (2023c) captures dynamic features by leveraging differences between frame-level and template features. DeepGaitV2-3D and DeepGaitV2-P3D Fan et al. (2023a) are variants of GaitBase Fan et al. (2023c) that utilize ordered input with 3D/P3D convolutions. VPNet Ma et al. (2024) employs a ResNet50-like backbone for gait recognition and introduces visual prompts to handle complex variations in gait patterns.

**Snippet Paradigm** We noticed that the term "snippet" has been previously used in the action recognition literature Wang et al. (2016); Duan et al. (2023), and we compare those approaches with our own. For instance, TSN Wang et al. (2016) constructs RGB snippets in a similar fashion but mandates that snippets be sampled from all segments and lacks intra-snippet modeling, which we consider crucial for snippet-based gait recognition. SkeleTR Duan et al. (2023) processes short skeleton sequences as snippets but requires continuity within each snippet. In our study, we extend the concept of snippets to gait recognition, where *neither the frames within a snippet nor the snippets in a sequence need to be strictly continuous*. Moreover, our approach diverges significantly from these methods by emphasizing snippet modeling, which will be elaborated in the next Section 3.2.

## 3 OUR APPROACH

In this work, we investigate a fundamental paradigm for gait recognition that addresses the limitations of unordered sets and ordered sequences. Specifically, we propose a new perspective that treats human gait as *a composition of individualized actions*, with each action represented by a **snippet**, which consists of a few frames randomly selected from a continuous segment of the sequence. This snippet paradigm allows the model to leverage both short-range and long-range temporal contexts during training, enhancing its capability for comprehensive gait feature learning.

In the following sections, we will first describe our strategy for organizing a sequence of silhouettes into snippets. Subsequently, we will present an effective approach to conduct snippet-based gait recognition.

### 3.1 SNIPPET SAMPLING

The underlying principles of sampling strategies for gait recognition can generally be summarized from two perspectives: (1) During training, a limited number of frames are typically sampled to represent a sequence due to the trade-off between computational cost and sampling diversity. (2) During inference, all frames of a sequence are utilized to ensure accurate recognition. Below, we briefly highlight the distinctions in sampling strategies when treating silhouettes as either unordered sets or ordered sequences. Specifically, in the training phase, set-based methods randomly select *discontinuous* frames from the entire sequence Chao et al. (2019), whereas sequence-based methods select *continuous* or *nearly continuous* frames for temporal modeling Fan et al. (2020).

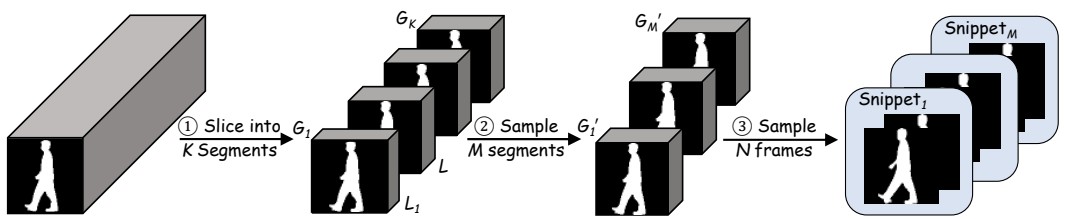

Figure 2: Snippet sampling for *training*. $\{G_1, \cdots, G_K\}$ represent the total segments of a sequence, where $L$ is the segment length and $L_1$ for the first segment is a random integer to enhance sampling diversity. $\{G'_1, \cdots, G'_M\}$ represent the sampled segments. $M$ and $N$ denote the number of sampled snippets per sequence and the number of sampled frames per snippet, respectively.

Our snippet-based sampling strategy influences both the training and inference phases, as described in detail in this section. It is noteworthy that we assume *the relative order of frames in a sequence is reliable, even though the frames themselves may not necessarily be continuous*, a condition that aligns well with practical applications Sepas-Moghaddam & Etemad (2021); Shen et al. (2022).

### 3.1.1 TRAINING PHASE

During the training phase, we first partition a sequence into non-overlapping segments of equal duration, preserving the relative order, and then design the snippet sampling strategy based on three guiding principles: (a) Given the constraints of computational resources and the need for sampling diversity, the total number of frames selected from a sequence should be limited, denoted as $S$. (b) The fundamental unit within the sampled $S$ frames is a snippet, where each snippet consists of $N$ frames randomly selected from a segment to capture an individualized action. (c) To increase sampling diversity and enhance model robustness, the segment partition for a sequence should vary across iterations.

Our approach is illustrated in Figure 2: (1) A sequence of silhouettes is divided into $K$ segments, denoted as $\{G_1, G_2, \cdots, G_K\}$, each of length $L$, where $L$ typically approximates the number of frames in a gait cycle (*e.g.*, $L = 16$ in most cases Ma et al. (2024)). If the sequence length is not perfectly divisible by $L$, the remaining frames are treated as an additional segment. (2) When processing a sequence in a mini-batch, we randomly sample $M$ segments from it and then randomly select $N$ frames from each chosen segment to construct the snippets. *Sampling with replacement* is allowed when the number of segments or the number of frames in a segment is limited. We ensure that $S = M \times N$, assigning each snippet a segment label $k$ ($k \in \{1, \cdots, K\}$) for subsequent modeling. (3) To enhance sampling diversity within a sequence, the initial frames are treated as a special segment, with its length $L_1$ randomly chosen from $\{1, 2, \cdots, L\}$.

### 3.1.2 INFERENCE PHASE

The snippet sampling strategy for the inference phase is also developed based on three guiding principles: (a) All frames in a sequence should be utilized to ensure precise matching between the probe and gallery. (b) To maintain consistency with the training phase, sequences are divided into segments, with all frames in each segment forming a snippet. (c) The segment partition should remain fixed to produce stable predictions.

Accordingly, our inference strategy involves the following three aspects: (1) A sequence of silhouettes is divided into $K$ segments of equal length $L$, as previously defined in the training phase (*e.g.*, $L = 16$). (2) *Each snippet comprises all frames within a segment*, and *prediction features are extracted using all snippets from the sequence*, which is equivalent to setting $M = K$ and $N = L$ during inference. (3) The length of the first segment $L_1$ is fixed to $L$, thereby eliminating the need for multiple forward passes and reducing inference overhead.

### 3.2 SNIPPET MODELING

Snippets provide a new paradigm for modeling silhouettes in gait recognition. However, fully exploiting the potential advantages of snippets remains an open question. In this work, we propose

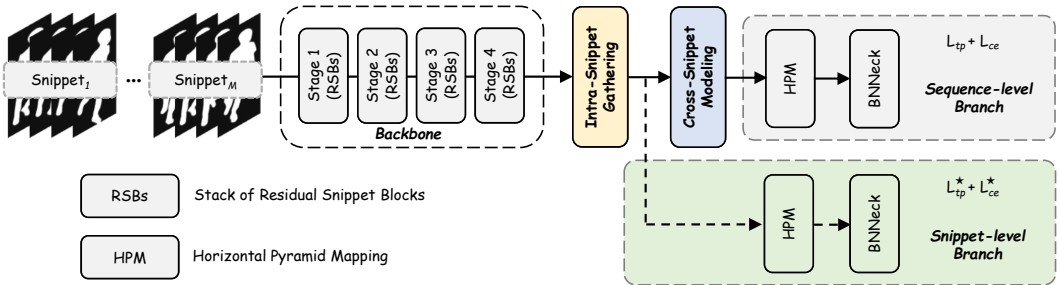

Figure 3: Illustration of GaitSnippet. (1) Residual Snippet Block *integrating Intra-Snippet Modeling* as shown in Figure 4(b) serves as the basic component to construct the backbone. (2) At the end of the backbone, we first apply *Intra-Snippet Gathering* (the *Gathering* step for Intra-Snippet Modeling) to derive snippet-level representations and then perform *Cross-Snippet Modeling* to obtain sequence-level representations. (3) In addition to sequence-level supervision, an auxiliary branch is introduced to enforce supervision on snippet-level features *only for training*.

an efficient yet effective solution to address this issue. Specifically, we identify three primary challenges in snippet modeling for gait recognition: **Intra-Snippet Modeling**, **Cross-Snippet Modeling**, and **Snippet-Level Supervision**. In the following sections, we systematically address these challenges through our proposed approach, which we term **GaitSnippet**. The pipeline is illustrated in Figure 3.

### 3.2.1 INTRA-SNIPPET MODELING

In GaitSnippet, we address intra-snippet modeling with the objective of *capturing local temporal context to enhance frame-level feature extraction* through a three-step process:

(1) *Gathering:* Considering that the frames within a snippet are not necessarily continuous, we treat a snippet as an unordered set. Based on this formulation, we utilize the efficient Set Pooling technique to aggregate the features of a snippet, which is implemented through a non-parametric Temporal Max Pooling operation Chao et al. (2019).

(2) *Smoothing:* To mitigate the negative effects of noise within a snippet and reduce the semantic gap between different levels of features, we apply a smoothing layer, typically implemented using a $1 \times 1$ convolution, following the *Gathering* step.

(3) *Residual:* To make frame-level feature extraction aware of local temporal context in a snippet, we incorporate a residual connection to merge the snippet-level output after smoothing with the frame-level features of the corresponding snippets.

As illustrated in Figure 4(a), these steps are formulated into a basic block called **Snippet Block**.

Furthermore, recent advancements in gait recognition have demonstrated that a plain 2D residual backbone Fan et al. (2023c;a) can achieve highly competitive performance in both constrained and unconstrained environments, while maintaining significantly lower computational costs compared to their 3D counterparts. The spatial convolution, specifically applied along the height and width dimensions, plays a critical role in extracting frame-level features. To facilitate effective collaboration between intra-snippet modeling and spatial convolution, we draw inspiration from P3D Qiu et al. (2017) and integrate a Snippet Block between two spatial convolutional layers within a standard residual block. The rationale behind this approach is to *enable each frame to become aware of local temporal context within a snippet during successive stages of frame-level feature extraction*. Ultimately, the architecture illustrated in Figure 4(b), called **Residual Snippet Block**, serves as the basic component to construct the backbone for GaitSnippet as shown in Figure 3.

### 3.2.2 CROSS-SNIPPET MODELING

For cross-snippet modeling, our objective is to *acquire a robust global representation for a gait sequence based on the snippet-level features*. As a pioneering attempt and to ensure a fair comparison with the base models Fan et al. (2023c;a), we conduct cross-snippet modeling on the output

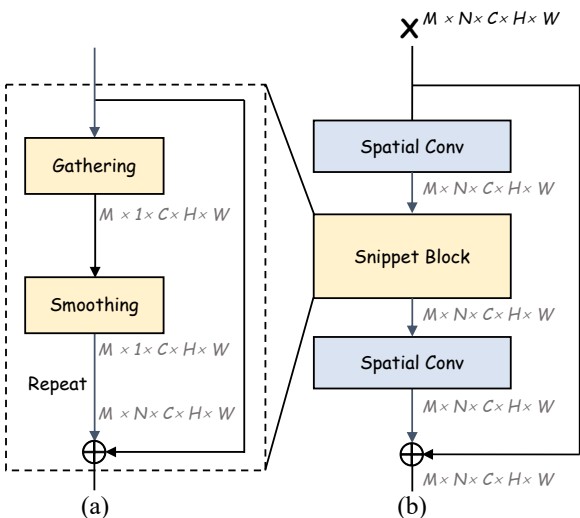

Figure 4: (a) Snippet Block. (b) Residual Snippet Block. $M$ and $N$ denote the number of snippets and the number of frames per snippet in a sequence, while $C$, $H$, and $W$ represent the dimensions of channel, height, and width.

of the backbone which corresponds to the *frame-level* features. Specifically, we first apply *Intra-Snippet Gathering* (the *Gathering* step for intra-snippet modeling) on the frame-level features to derive snippet-level representations. Subsequently, we treat all snippets from a sequence as an unordered set and employ another Set Pooling Chao et al. (2019) to perform cross-snippet modeling. In practice, this is implemented using Temporal Max Pooling on all *snippet-level* representations within a sequence.

It is crucial to highlight that (1) GaitSnippet involves two *hierarchical* unordered sets: frames within a snippet and all snippets within a sequence. However, the snippet-based modeling approach is *not* permutation-invariant to the frame order, distinguishing it from methods that exclusively rely on unordered sets Chao et al. (2019); Fan et al. (2023c). Unlike unordered sets, the use of snippets enables the exploitation of local temporal context in frame-level feature extraction, which is vital for learning discriminative and complementary features for individual silhouettes. (2) At the end of the backbone, Temporal Max Pooling employed for both intra-snippet and cross-snippet modeling makes the sequence-level features equivalent to the maximum of all frames. Yet the intermediate output from intra-snippet modeling is essential for enabling Snippet-Level Supervision. Further discussion about the role of Temporal Max Pooling is provided in Section A.4.2 of the appendix.

### 3.2.3 SNIPPET-LEVEL SUPERVISION

The snippet-based modeling of gait conveniently facilitates the extraction of two hierarchical representations for a sequence, namely, *sequence-level* and *snippet-level* representations. For supervision on the *sequence-level representations*, we adopt the typical approach outlined in Fan et al. (2023c). Initially, Horizontal Pyramid Mapping Fu et al. (2019); Chao et al. (2019) (including linear layers for separate parts) is utilized to horizontally split the features for obtaining fine-grained part representations efficiently. Then, for each part, we employ triplet loss $\mathcal{L}_{tp}$ and cross-entropy loss $\mathcal{L}_{ce}$, assisted by BNNeck Luo et al. (2019), for training. Formally, these losses are defined as follows:

$$\mathcal{L}_{tp} = \frac{1}{N_{tp}} \overbrace{\sum_{u=1}^{U} \sum_{v=1}^{V}}^{anchor} \overbrace{\sum_{a=1}^{V}}^{pos} \overbrace{\sum_{\substack{b=1 \\ b \neq u}}^{U} \sum_{c=1}^{V}}^{neg} [\delta + \mathcal{D}(\mathcal{F}_{u,v}, \mathcal{F}_{u,a}) - \mathcal{D}(\mathcal{F}_{u,v}, \mathcal{F}_{b,c})]_+$$

$$\mathcal{L}_{ce} = -\frac{1}{U \times V} \overbrace{\sum_{u=1}^{U} \sum_{v=1}^{V}}^{batch} \overbrace{\sum_{c=1}^{N_c}}^{sub} q_{u,v,c} \log p_{u,v,c}$$

(1)

Here, *pos*, *neg*, and *sub* stand for *positive*, *negative*, and *subjects*, respectively. $(U, V)$ denote the number of subjects and the number of sequences per subject in a mini-batch. $N_{tp}$ serves as a normalization coefficient accounting for the non-zero triplet terms. $\delta$ is a margin threshold and $[\ ]_+$ works as the ReLU function. $\mathcal{F}$ denotes the sequence-level representations and $\mathcal{D}$ measures the Euclidean distance. $(\mathcal{F}_{u,v}, \mathcal{F}_{u,a})$ and $(\mathcal{F}_{u,v}, \mathcal{F}_{b,c})$ represent positive and negative pairs, respectively. $N_c$ is the number of subjects in the training set, while $p$ and $q$ denote the predicted probabilities and the one-hot ground-truth identity labels.

With the snippet-based approach to gait, we can conveniently obtain snippet-level representations in addition to sequence-level representations, motivating us to introduce the following fine-grained supervision. Specifically, we add a separate branch to process snippet-level representations prior to cross-snippet modeling, using Horizontal Pyramid Mapping to obtain part-level features and incorporating BNNeck analogous to the sequence-level branch. Formally, for each part, the snippet-level triplet loss $\mathcal{L}_{tp}^{\star}$ and cross-entropy loss $\mathcal{L}_{ce}^{\star}$ are computed as follows:

$$\mathcal{L}_{tp}^{\star} = \frac{1}{N_{tp}^{\star}} \overbrace{\sum_{u=1}^{U} \sum_{v=1}^{V}}^{batch} \overbrace{\sum_{m=1}^{M}}^{snp} \overbrace{\sum_{a=1}^{V}}^{pos} \overbrace{\sum_{i=1}^{M}}^{snp} \overbrace{\sum_{\substack{b=1 \\ b \neq u}}^{U} \sum_{c=1}^{V}}^{neg} \overbrace{\sum_{j=1}^{M}}^{snp} \left[\delta + \mathcal{D}(\mathcal{F}_{u,v,m}^{\star}, \mathcal{F}_{u,a,i}^{\star}) - \mathcal{D}(\mathcal{F}_{u,v,m}^{\star}, \mathcal{F}_{b,c,j}^{\star})\right]_+ \tag{2}$$

$$\mathcal{L}_{ce}^{\star} = -\frac{1}{U \times V \times M} \overbrace{\sum_{u=1}^{U} \sum_{v=1}^{V}}^{batch} \overbrace{\sum_{m=1}^{M}}^{snp} \overbrace{\sum_{c=1}^{N_c}}^{sub} q_{u,v,m,c}^{\star} \log p_{u,v,m,c}^{\star}$$

where *snp* denotes snippets, $M$ is the number of sampled snippets per sequence for training, and $\mathcal{F}^{\star}$ refers to snippet-level representations. The remaining symbols are similar to those in Eq. 1, with the superscript $\star$ indicating snippet-level computations.

We then define the integrated objective for one of the part representations as:

$$\mathcal{L}_{all} = \mathcal{L}_{tp} + \mathcal{L}_{ce} + \alpha \times (\mathcal{L}_{tp}^{\star} + \mathcal{L}_{ce}^{\star}) \tag{3}$$

where $\alpha$ is a hyperparameter to balance the two levels of supervision signals. The final loss is computed by averaging the above losses across all parts, which is used to train the entire network.

It is important to emphasize that the additional branch for snippet-level supervision is employed *only during the training phase*, thereby leaving the inference complexity unaffected. For evaluation, we utilize the features extracted before BNNeck in the sequence-level branch to compute similarities between the probe and gallery sequences.

## 4 EXPERIMENTS

### 4.1 SETTINGS

We conduct experiments on four widely-used gait datasets: Gait3D Zheng et al. (2022) and GREW Zhu et al. (2021), CCPG Li et al. (2023) and CCGR-MINI Zou et al. (2024). In the training phase, we adopt $L = 16$ to approximate the number of frames depicting a gait cycle Ma et al. (2024) for segment partition, and $L_1$ is a random integer sampled from $\{1, 2, \cdots, 16\}$. To sample a sequence, we randomly select $M = 4$ snippets and $N = 8$ frames per snippet, *i.e.*, we sample $S = 32$ frames for each sequence. For evaluation, we set $L_1 = L = 16$ for segment partition. All frames in a segment are treated as a snippet, and all snippets for a sequence are used to extract gait features. Detailed dataset statistics and implementation details are provided in the appendix.

### 4.2 PERFORMANCE COMPARISON

**Gait3D & GREW** The emergence of Gait3D and GREW has advanced gait recognition research from controlled laboratory settings to real-world environments. In Table 1, we present a performance comparison on in-the-wild benchmarks. The methods are categorized into three groups based on how they treat the input: *Ordered Sequences*, *Unordered Sets*, and the brand-new *Snippets*.

| Method | Cate-gory | Back-bone | Gait3D | | GREW | |
|---|---|---|---|---|---|---|
| | | | R1 | mAP | R1 | R5 |
| GaitPart Fan et al. (2020) | | 2D | 28.2 | 21.6 | 47.6 | 60.7 |
| GaitGL Lin et al. (2021) | | 3D | 29.7 | 22.3 | 47.3 | 63.6 |
| GaitGCI Dou et al. (2023) | | 3D | 50.3 | 39.5 | 68.5 | 80.8 |
| DyGait Wang et al. (2023c) | | 3D | 66.3 | 56.4 | 71.4 | 83.2 |
| HSTL Wang et al. (2023a) | | 3D | 61.3 | 55.5 | 62.7 | 76.6 |
| SwinGait-3D Fan et al. (2023a) | *Seq* | Swin3D | 75.0 | **67.2** | 79.3 | 88.9 |
| DeepGaitV2-3D Fan et al. (2023a) | | 3D | 72.8 | 63.9 | 79.4 | 88.9 |
| DeepGaitV2-P3D Fan et al. (2023a) | | P3D | 74.4 | 65.8 | 77.7 | 87.9 |
| VPNet Ma et al. (2024) | | 3D | **75.4** | / | **80.0** | **89.4** |
| CLTD Xiong et al. (2024) | | 3D | 69.7 | / | 78.0 | 87.8 |
| GaitMoE Huang et al. (2024) | | 3D | 73.7 | 66.2 | 79.6 | 89.1 |
| GaitSet Chao et al. (2019) | | 2D | 36.7 | 30.0 | 48.4 | 63.6 |
| GaitBase Fan et al. (2023c) | *Set* | 2D | 64.6 | 55.3 | 60.1 | 75.5 |
| SwinGait-2D Fan et al. (2023a) | | Swin2D | **69.4** | **61.6** | **70.8** | **83.7** |
| DeepGaitV2-2D Fan et al. (2023a) | | 2D | 68.2 | 60.4 | 68.6 | 82.0 |
| GaitSnippet (**Ours**) | *Snippet* | 2D | **77.5** | **69.4** | **81.7** | **90.9** |

Table 1: Performance comparison on Gait3D Zheng et al. (2022) and GREW Zhu et al. (2021). The results are reported in rank-1 (R1, %), rank-5 (R5, %), and mean Average Precision (mAP, %). The best results in each category are marked in **red**, **blue**, and **bold**, respectively.

From the results in Table 1, the following observations can be made: (1) Sequence-based methods achieve state-of-the-art performance and mostly employ 3D or P3D convolution in the backbone, which generally entails higher computational costs compared to 2D convolution-based backbones. (2) DeepGaitV2-2D Fan et al. (2023a), despite their simplicity, achieve highly competitive performance on these benchmarks. (3) GaitSnippet outperforms advanced methods on both benchmarks using a 2D convolutional backbone. Specifically, the performance gains compared to 2D convolution-based baselines (*e.g.*, R1: +9.3%, mAP: +9.0% over DeepGaitV2-2D on Gait3D with the same network depth) effectively demonstrate the effectiveness of snippet-based gait recognition.

**CCPG & CCGR-MINI** With the increasing interest in gait recognition, several new datasets have recently been introduced Li et al. (2023); Shen et al. (2023); Li et al. (2024); Zou et al. (2024), aiming to address more diverse and challenging scenarios. To further demonstrate the generalizability of GaitSnippet, we additionally evaluate it on two representative emerging datasets: CCPG Li et al. (2023)[1] and CCGR-MINI Zou et al. (2024). As shown in Table 2, GaitSnippet achieves state-of-the-art performance on both datasets, further validating the effectiveness and adaptability of snippet-based modeling for gait recognition.

## 4.3 ABLATION STUDY

### 4.3.1 ABLATION STUDY ON SNIPPET SAMPLING

In Table 3, we present an ablation study on Snippet Sampling from two perspectives.

First, we evaluate the overall effect of the sampling strategy using our base model (*i.e.*, DeepGaitV2-2D) in the first part, including set-based, sequence-based, and snippet-based strategies. Interestingly, from the first three rows, we observe that Snippet Sampling during training also benefits recognition performance with DeepGaitV2-2D, which is based on unordered sets. A likely reason for this is that our sampling strategy enhances the robustness of DeepGaitV2-2D by narrowing the distribution gap between discontinuous frames during training and continuous sequences during testing.

Second, we analyze the effect of hyper-parameters for Snippet Sampling in the second part. Specifically, during the training phase, there are four hyper-parameters for sampling snippets from a sequence: $L$–the segment length, $S$–the total number of frames sampled from a sequence, $M$–the number of snippets sampled from a sequence, and $N$–the number of frames sampled per snippet from a segment. Note that $S = M \times N$ is always maintained. During our experiments with Snippet Sampling, we fix $S = 32$, considering computational cost and ensuring fair comparisons. In Table 3,

---

[1]We re-ran the experiments on CCPG using the same preprocessing and protocols as the baselines Fan et al. (2023a;c) to ensure fair comparisons. The results differ slightly from those in the initial submission, but they remain state of the art.

| Method | Category | Backbone | CCPG | | | | | CCGR-MINI | |
|---|---|---|---|---|---|---|---|---|---|
| | | | CL | UP | DN | BG | AVG | R1 | mAP |
| GaitPart Fan et al. (2020) | *Seq* | 2D | 79.2 | 85.3 | 86.5 | 88.0 | 84.8 | 8.0 | 10.1 |
| DeepGaitV2-P3D Fan et al. (2023a) | | P3D | **90.5** | **96.3** | **91.4** | **96.7** | **93.7** | **39.4** | **36.0** |
| GaitSet Chao et al. (2019) | *Set* | 2D | 77.5 | 85.0 | 82.9 | 87.5 | 83.2 | 13.8 | 15.4 |
| GaitBase Fan et al. (2023c) | | 2D | **88.5** | **92.7** | **93.4** | **93.2** | **92.0** | **27.0** | **24.9** |
| GaitSnippet (**Ours**) | ***Snippet*** | 2D | 91.5 | 96.6 | 94.6 | 97.7 | 95.1 | 42.4 | 39.5 |

Table 2: Performance comparison on CCPG Li et al. (2023) and CCGR-MINI Zou et al. (2024). The results on CCPG are reported in rank-1 (R1, %) accuracy, while those on CCGR-MINI are reported in rank-1 accuracy (R1, %) and mean Average Precision (mAP, %). CL, UP, DN, BG, and AVG refer to changing full outfits, changing top clothes, changing pants, walking with bags, and mean accuracy, respectively. The best performance in each category is highlighted in **red**, **blue**, and **bold**, respectively.

we conduct ablation studies on the other sampling parameters. (1) We set $L = 16$ to approximate the number of frames in a gait cycle Ma et al. (2024), and we also tried $L \in \{8, 32\}$ in the fourth/fifth rows. (2) We set $N = 8$, which is half of a gait cycle, and also tried $N \in \{4, 16\}$ in the sixth/seventh rows. (3) Given that $S = M \times N$, $M$ varies with different values of $N$ in each case (*i.e.*, $M = 4$ for $N = 8$, $M = 8$ for $N = 4$, and $M = 2$ for $N = 16$).

### 4.3.2 ABLATION STUDY ON SNIPPET BLOCK

In this section, we analyze the three steps involved in intra-snippet modeling, namely *Gathering*, *Smoothing*, and *Residual*. The results are presented in the first part of Table 4.

To clarify the results: (1) If none of the techniques in Table 4 is applied, GaitSnippet is equivalent to DeepGaitV2-2D with Snippet Sampling. (2) In the first row, when *Gathering* is removed for each stage of intra-snippet modeling, snippet-level supervision becomes inapplicable. (3) In the second row, the experiment highlights the importance of the smoothing layer, which acts as a bridge between frame-level features and those aggregated from a snippet. When *Smoothing* is removed, *the model for inference does not introduce any additional parameters*, as the snippet-level branch is only employed during training. In this case, the local context modeling within each Snippet Block still works but is more susceptible to the noise within a snippet and the semantic gap between different levels of features. (4) For the third row, it is worth emphasizing that *Residual* in Table 4 refers to *the integration of local contextual information from a snippet with frame-level features as depicted in Figure 4(a)*, rather than the standard residual connection shown in Figure 4(b). When *Residual* is removed, only the contextual information from the snippet is used for subsequent layers, which inevitably results in the loss of fine-grained details from each silhouette after the first Snippet Block.

### 4.3.3 ABLATION STUDY ON SNIPPET-LEVEL SUPERVISION

Benefitting from the snippet paradigm, we can easily incorporate snippet-level losses to provide fine-grained supervision for gait feature learning. In the second part of Table 4, we perform ablation studies to analyze the effect of snippet-level supervision. We can observe that: (1) When $\alpha = 0$, as shown in the fifth row, only the sequence-level loss is used to train the entire model, and our approach still achieves highly competitive performance. (2) Snippet-level supervision improves recognition performance with varied loss weights.

Additionally, in the last row of Table 4, we experiment with sharing weights between the sequence-level and snippet-level branches shown in Figure 3, consisting of Horizontal Pyramid Mapping and BNNeck. The corresponding results show a moderate performance degradation, likely due to the semantic gap between the sequence-level and snippet-level features, making weight sharing between the branches inappropriate.

## 5 CONCLUSION

In this work, we explore a new paradigm for gait recognition that integrates the strengths of unordered sets and ordered sequences. Motivated by the observation that human identification does not necessarily rely on a complete gait cycle Giese & Poggio (2003), we conceptualize human gait as a

| Model | Sampling Strategy | $L$ | $M$ | $N$ | R1 | mAP |
|---|---|---|---|---|---|---|
| DeepGaitV2-2D | *Set* | - | - | - | 68.2 | 60.4 |
| | *Seq* | - | - | - | 66.0 | 58.7 |
| | *Snippet* | 16 | 4 | 8 | 69.5 | 61.5 |
| GaitSnippet | *Snippet* | 8 | 4 | 8 | 76.4 | 69.0 |
| | | 32 | 4 | 8 | 74.7 | 67.5 |
| | | 16 | 8 | 4 | 74.3 | 66.7 |
| | | 16 | 2 | 16 | 75.2 | 66.9 |
| | | **16** | **4** | **8** | **77.5** | **69.4** |

Table 3: Ablation study on snippet sampling. $L$, $M$, and $N$ denote the segment length for sequence partition, the number of snippets sampled per sequence, and the number of frames sampled per snippet, respectively. The results are reported on Gait3D.

| Snippet Block | | | Snippet Supervision | | R1 | mAP |
|---|---|---|---|---|---|---|
| *Gathering* | *Smoothing* | *Residual* | $\alpha$ | *SW* | | |
| × | √ | √ | - | - | 73.3 | 65.7 |
| √ | × | √ | 0.75 | × | 74.8 | 66.6 |
| √ | √ | × | 0.75 | × | 72.5 | 63.7 |
| √ | √ | √ | **0.75** | × | **77.5** | **69.4** |
| √ | √ | √ | 0.00 | × | 75.8 | 68.5 |
| √ | √ | √ | 0.50 | × | 76.4 | 69.4 |
| √ | √ | √ | 1.00 | × | 76.6 | 69.4 |
| √ | √ | √ | 0.75 | √ | 75.5 | 68.8 |

Table 4: Ablation study on snippet modeling on Gait3D. *Gathering*, *Smoothing*, and *Residual* are the three steps in a Snippet Block. $\alpha$ represents the loss weight for snippet-level losses, and *SW* denotes sharing weights between the sequence-level and snippet-level branches in Figure 3. The experiments are conducted on Gait3D.

combination of individualized actions, with each action represented by a few frames that are adjacent but not necessarily continuous. In essence, gait snippets enable the model to simultaneously exploit both short-range and long-range temporal contexts, which is beneficial for learning identity-related features from entire walking sequences. Furthermore, we provide a non-trivial solution based on gait snippets, addressing the challenges of Snippet Sampling and Snippet Modeling. Extensive experiments across various benchmarks demonstrate that our approach consistently improves performance and achieves state-of-the-art results in the wild, effectively verifying the potential of snippet-based gait recognition.

## ETHICAL STATEMENT

The datasets used in our experiments are widely adopted in the literature, with informed consent obtained from all subjects during data collection. Additionally, no personal identifiers are accessible. We strongly advocate that research in this field should be conducted with strict privacy protection measures in place.

## REPRODUCIBILITY STATEMENT

All experiments are conducted on publicly available gait recognition datasets (Gait3D, GREW, CCPG and CCGR-MINI) following the official splits and evaluation protocols. Detailed hyper-parameters and training procedures are provided in the paper. The complete code and pre-trained models will be released.

## ACKNOWLEDGEMENT

This work is jointly supported by National Natural Science Foundation of China (62276025, 62476027) and the Fundamental Research Funds for the Central Universities (2253200026).

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
