| Dataset | Train Set | | Test Set | | Walking Condition | #Cam |
|---------|-----------|--|----------|--|-------------------|------|
|         | #ID | #Seq | #ID | #Seq | | |
| Gait3D | 3000 | 18940 | 1000 | 6369 | Diverse | 39 |
| GREW | 20000 | 102887 | 6000 | 24000 | Diverse | 882 |
| CCPG | 100 | 8187 | 100 | 8379 | NM/BG/CL | 10 |
| CCGR-MINI | 570 | 27507 | 400 | 20377 | Diverse | 33 |

Table 5: Dataset statistics. For each dataset, we present the number of subjects (#ID) and sequences (#Seq), walking conditions (NM/BG/CL for normal walking, walking with bags, and walking in different clothes), and the number of cameras (#Cam).

| Dataset | *blocks* | *channels* | *strides* |
|---------|----------|------------|-----------|
| Gait3D | [1, 4, 4, 1] | [64, 128, 256, 512] | [1, 2, 2, 1] |
| GREW | [3, 4, 6, 3] | [64, 128, 256, 512] | [1, 2, 2, 1] |
| CCPG | [1, 1, 1, 1] | [64, 128, 256, 512] | [1, 2, 2, 1] |
| CCGR-MINI | [1, 4, 4, 1] | [64, 128, 256, 512] | [1, 2, 2, 1] |

Table 6: The backbone settings for each dataset. *Blocks*, *channels*, and *strides* refer to the number of blocks, convolutional channels, and strides for all stages, respectively. We configure the snippet-based backbone for each dataset with reference to the sequence-based counterparts Fan et al. (2023a); Ma et al. (2024).

# A APPENDIX

## A.1 DATASET DETAILS

The statistics for the widely-used datasets employed in our research, namely Gait3D Zheng et al. (2022), GREW Zhu et al. (2021), CCPG Li et al. (2023) and CCGR-MINI Zou et al. (2024), are presented in Table 5.

**Gait3D** is a large-scale benchmark dataset captured in a supermarket environment, with two two-hour video segments randomly selected from each of seven days. During evaluation, one sequence per subject is designated as the probe, while the remaining sequences are utilized as the gallery.

**GREW** is collected from multiple cameras in an uncontrolled environment over the course of a single day, resulting in diverse view variations. Following the official evaluation protocol, each subject has four sequences, with two sequences used as the probe and the remaining two as the gallery.

**CCPG** is a cloth-changing benchmark dataset for person re-identification and gait recognition. It includes sequences of subjects captured in indoor and outdoor scenes, with the subjects having different clothing variations. Following the standard protocol, the subjects are divided into two parts: the first half is used for training, and the remaining data is used for testing.

**CCGR-MINI** is a subset of the Cross-Covariate Gait Recognition (CCGR) dataset, specifically designed to address covariate diversity at both population and individual levels. CCGR-MINI retains the diversity while enabling efficient evaluation and training under limited computational budgets.

## A.2 IMPLEMENTATION DETAILS

Residual Snippet Block in Figure 4(b) is used as the basic component to construct the backbone for GaitSnippet, where the smoothing layer is implemented using a $1 \times 1$ convolution. The number of blocks (*blocks*), convolutional channels (*channels*), and strides (*strides*) for each stage across the four datasets are detailed in Table 6, referring to the network configurations used in Fan et al. (2023a); Ma et al. (2024), *e.g.*, *blocks* = $[1, 4, 4, 1]$ for DeepGaitV2 on Gait3D Fan et al. (2023a) and *blocks* = $[3, 4, 6, 3]$ for VPNet on GREW Ma et al. (2024). The settings for CCPG and CCGR-MINI follow those used for Gait3D, with one modification: the backbone architecture used for CCPG employs a reduced number of *blocks*, specifically $[1, 1, 1, 1]$.

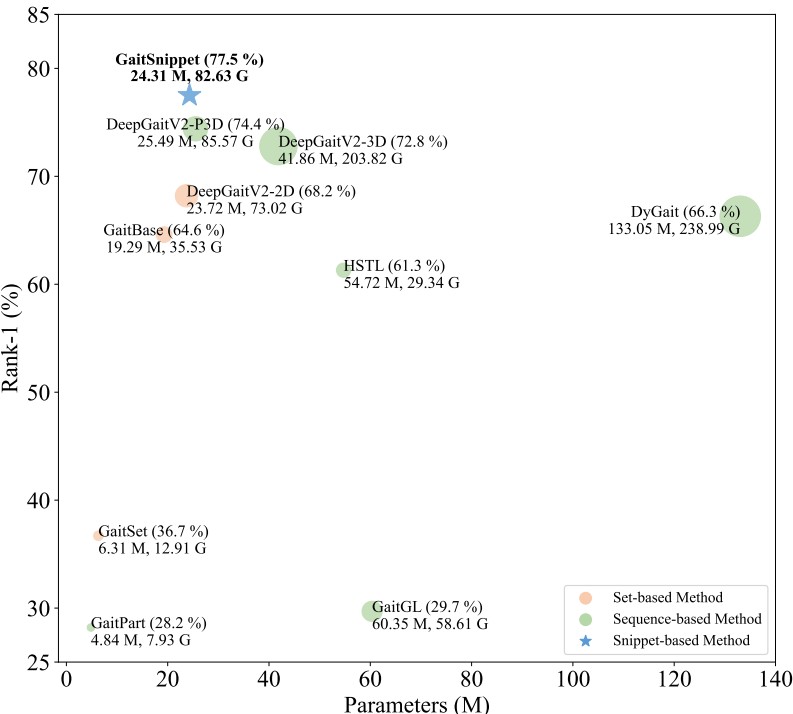

Figure 5: Computation cost in terms of parameters and FLOPs. The statistics are obtained on Gait3D, following the methodology of Wang et al. (2023b); Huang et al. (2024).

Besides, the margin threshold $\delta$ for triplet loss is set to $0.2$ and the loss weight $\alpha$ for snippet-level supervision is set to $0.75$. During inference, *Snippet-level Branch* is disabled, and we use *Sequence-level Branch* output for similarity computation. The feature dimensions match DeepGaitV2 (*e.g.*, $16 \times 256$ on Gait3D/GREW). For other settings, such as data preprocessing and training strategies, we refer to those described in Fan et al. (2023a); Ma et al. (2024). To ensure reproducibility, the PyTorch-based source code and pretrained models will be made publicly available.

## A.3 MORE EXPERIMENTAL RESULTS

### A.3.1 ANALYSES ON COMPUTATION COST

In Figure 5, we compare the computational cost of GaitSnippet with several representative methods in terms of parameter count and FLOPs. The statistics are obtained on the Gait3D dataset, following the methodology of Wang et al. (2023b); Huang et al. (2024). We focus our comparison on DeepGaitV2-2D/3D/P3D, all of which adopt the same network depth on Gait3D.

(1) Notably, GaitSnippet has significantly fewer parameters and FLOPs than DeepGaitV2-3D, and even fewer than DeepGaitV2-P3D.

(2) Compared to DeepGaitV2-2D, GaitSnippet exhibits a slightly higher computational cost, primarily due to the introduction of smoothing layers and temporal aggregation in intra-snippet modeling. However, this modest increase is justified by a substantial performance improvement over DeepGaitV2-2D, with a +9.3% gain in Rank-1 accuracy and a +9.0% gain in mAP on Gait3D. It is noteworthy that GaitSnippet also outperforms both DeepGaitV2-3D and DeepGaitV2-P3D.

(3) To rule out the impact of model size, we further evaluate a lightweight version of GaitSnippet with reduced network depth (*blocks* = $[1, 3, 3, 1]$), resulting in only 22.8M parameters and 68.6G FLOPs lower than those of DeepGaitV2-2D. Despite this compact design, the model achieves competitive performance (Rank-1: 77.0%, mAP: 69.3%), still outperforming DeepGaitV2-2D and even DeepGaitV2-3D/P3D. This confirms that the performance gains stem from the proposed snippet-based modeling rather than increased model complexity.

| Feature Type | $L_1$ (*for evaluation*) | R1 | mAP |
|---|---|---|---|
| *Sequence-level Features* | 16 | 77.5 | 69.4 |
| *Sequence-level Features* | 8 | 77.3 | 69.5 |
| *Sequence-level Features* | $\{8, 16\}$ | 77.2 | 69.8 |
| *Snippet-level Features* | 16 | 65.2 | 54.8 |

Table 7: Ablation study on evaluation. *Sequence-level Features* and *Snippet-level Features* denote the features before BNNeck generated by the sequence-level and snippet-level branches, respectively. $L_1$ denotes the length of the first segment *for evaluation*. For the ensemble of different $L_1$ values, we average the features from multiple sequence partitions. The results are reported on Gait3D in terms of rank-1 accuracy (R1, %) and mean Average Precision (mAP, %).

| Model | R1 | mAP |
|---|---|---|
| GaitSet Chao et al. (2019) | 36.7 | 30.0 |
| GaitSet Chao et al. (2019) + Snippet | **48.2** | **39.1** |
| GaitBase Fan et al. (2020) | 64.6 | 55.3 |
| GaitBase Fan et al. (2020) + Snippet | **69.7** | **60.2** |

Table 8: Ablation study on generalization. Experiments are conducted on Gait3D, with results reported in terms of rank-1 accuracy (R1, %) and mean Average Precision (mAP, %).

### A.3.2 ABLATION STUDY ON EVALUATION

During the evaluation phase, we fix $L_1 = 16$, which requires only a single forward pass and does not add additional inference burden. In this section, we explore different values of $L_1$ (*e.g.*, $L_1 = 8$) and evaluate the ensemble of multiple segment partitions (*e.g.*, $L_1 \in \{8, 16\}$) *only for evaluation*. The results in Table 7 indicate that: (1) The segment partition has a minor effect on evaluation, as the model is trained with various partition strategies. (2) Averaging the features from multiple segment partitions can slightly improve recognition performance in terms of mAP, but this significantly increases the inference cost. Further discussion is provided in Section A.4.2.

Additionally, in the last row of Table 7, we experiment with using snippet-level representations for evaluation. Snippet-level features from a sequence are averaged to match the probe and gallery. Unsurprisingly, the results are inferior to those using sequence-level representations which effectively capture long-range temporal dependencies.

### A.3.3 ABLATION STUDY ON GENERALIZATION

In this section, we aim to verify the generalization capability of our approach by applying the snippet paradigm to some set-based methods. It is important to note that sequence-based methods rely on *continuous* input, whereas both the snippets within a sequence and the frames within each snippet are *discontinuous*, making it infeasible to transform sequence-based methods into their snippet-based counterparts. Thus, we adopt the set-based methods GaitSet Chao et al. (2019) and GaitBase Fan et al. (2023c) for our analysis. Specifically: (1) We redesign the input phase using the snippet-based sampling strategy. (2) We insert a Snippet Block, as illustrated in Figure 4, between two convolutional layers in each stage of the backbone. Finally, we observe significant performance gains, as shown in Table 8.

### A.3.4 ABLATION STUDY ON FRAME SAMPLING

In Section 3.1, we provide a brief review of the sampling strategies used in recent set-based and sequence-based studies Chao et al. (2019); Fan et al. (2020); Lin et al. (2021). Typically, a limited number of frames (*i.e.*, $S = 30$ in most cases) are sampled during training for each sequence. In our experiments, however, we sample $S = 32$ frames per sequence, corresponding to $M = 4$ snippets per sequence and $N = 8$ frames per snippet. For a rigorous ablation study, we also apply a sampling strategy of $S = 32$ frames per sequence to several representative baseline methods Fan et al. (2023a), which has a slight impact on performance, as indicated in Table 9.

| Model | Sampling Strategy | $S$ | R1 | mAP |
|---|---|---|---|---|
| DeepGaitV2-2D Fan et al. (2023a) | *Set* | $\min:10, \max:50$ | 68.2 | 60.4 |
| DeepGaitV2-3D Fan et al. (2023a) | *Seq* | 30 | 72.8 | 63.9 |
| DeepGaitV2-P3D Fan et al. (2023a) | *Seq* | 30 | 74.4 | 65.8 |
| DeepGaitV2-2D Fan et al. (2023a) | *Set* | 32 | 68.6 | 60.1 |
| DeepGaitV2-3D Fan et al. (2023a) | *Seq* | 32 | 72.1 | 64.6 |
| DeepGaitV2-P3D Fan et al. (2023a) | *Seq* | 32 | 74.2 | 65.8 |

Table 9: Ablation study on frame sampling. *Set* and *Seq* represent the set-based Chao et al. (2019) and sequence-based Fan et al. (2020) sampling strategies, respectively. $S$ denotes the number of frames sampled per sequence during the training phase. In the original implementation of DeepGaitV2-2D Fan et al. (2023a), $S$ is randomly selected from $\{10, 11, \cdots, 49, 50\}$. We conduct the experiments on Gait3D and report the results in terms of rank-1 accuracy (R1, %) and mean Average Precision (mAP, %).

### A.3.5 SNIPPETS ON MISSING FRAMES

In GaitSnippet, we partition each trajectory into contiguous segments and randomly sample frames within each segment to form snippets. This reduces reliance on "perfect" gait cycles and emphasizes recurring cues that are robust to frame dropping.

To further validate this, we simulate different frame-dropping ratios at test time on Gait3D and report rank-1 accuracy in Table 10. GaitSnippet consistently outperforms DeepGaitV2-P3D, and the margin increases as fewer frames are retained.

### A.3.6 SNIPPETS ON SKELETON MAPS

Our snippet paradigm can also be applied to skeleton maps Fan et al. (2024). We conduct additional experiments by directly transferring snippet-based modeling to the skeleton-map setting, as shown in Table 11:

(1) **GaitSnippet (SkeletonMap)**: we simply replace silhouette inputs with skeleton maps, while keeping the snippet sampling strategy and Snippet Blocks unchanged.

(2) **GaitSnippet (Silhouettes+SkeletonMap)**: a multi-modal extension that follows the fusion strategy of SkeletonGait++, but replaces both the silhouette and skeleton branches with our snippet-based counterparts.

In these experiments, both variants consistently outperform SkeletonGait and SkeletonGait++, respectively. This empirically demonstrates that i) the snippet paradigm is *not tied to silhouettes* but serves as a general temporal modeling scheme that can be plugged into skeleton-map-based architectures; and ii) even in the multi-modal setting, snippet-based modeling brings further gains.

### A.4 MORE DISCUSSION

### A.4.1 DISCUSSION ON SNIPPET SAMPLING

In this section, we provide additional clarifications on Snippet Sampling from three perspectives:

(1) *Frame Order for Segment Partition*. As clarified in Section 3.1, the snippet-based sampling relies on frame order to partition the sequence into segments. As a result, the snippet-based modeling is *not* permutation invariant to frame order, making it inappropriate to categorize it into the set-based category Zaheer et al. (2017). In our formulation, frames in a snippet are randomly sampled from a continuous segment of the sequence to describe an action, which forms the basis for crucial local context modeling.

(2) *Random Selection within Each Segment*. Randomly selecting frames within each segment for snippet sampling reduces dependency on continuous input and enhances robustness to missing silhouettes. An interesting future direction could involve sampling snippets by identifying important silhouettes Hou et al. (2022b); Wang et al. (2024); however, measuring frame importance is challenging, especially during the input phase.

| Reserved Frame Ratio | 1/1 | 1/2 | 1/3 | 1/4 |
|---|---|---|---|---|
| DeepGaitV2-P3D Fan et al. (2023a) | 74.4 | 69.6 | 63.5 | 57.1 |
| GaitSnippet | **77.5** | **73.7** | **70.3** | **66.2** |
| Improvement | **+3.1** | **+4.1** | **+6.8** | **+9.1** |

Table 10: Snippets on missing frames. We conduct experiments on Gait3D and report rank-1 accuracy (R1, %).

| Method | R1 | mAP |
|---|---|---|
| SkeletonGait Fan et al. (2024) | 38.1 | 28.9 |
| GaitSnippet (SkeletonMap) | **40.3** | **31.0** |
| SkeletonGait++ Fan et al. (2024) | 77.6 | 70.3 |
| GaitSnippet (Silhouettes+SkeletonMap) | **78.8** | **73.6** |

Table 11: Snippets on skeleton maps. Experiments are conducted on Gait3D, with results reported in terms of rank-1 accuracy (R1, %) and mean Average Precision (mAP, %).

(3) *Data Augmentation*. The augmentation strategy of varying the first segment to increase snippet diversity is naturally suited for snippet-based gait recognition. Similarly, state-of-the-art set-based and sequence-based methods adopt their own specific augmentation strategies, such as randomly sampling frames from a continuous segment in sequence-based methods Fan et al. (2020); Wang et al. (2023a). As evidenced in previous works, such augmentation does not hinder fair comparisons under the same evaluation protocol Hou et al. (2022a).

(4) *Sampling Hyper-parameters*. During training, *sampling with replacement* is adopted when the sequence length is insufficient. If the number of segments $K$ is smaller than $M$, some segments may be sampled multiple times. Similarly, if a segment contains fewer than $N$ frames, repeated frames may appear within a snippet. During inference, all segments are used ($M = K$), and each snippet includes all frames in the segment ($N = L$).

It is worth noting that, the segment length $L$ is an important hyper-parameter for both training and inference, and $K$ is derived from the sequence length and $L$. We set $L = 16$ as an empirical approximation of the frames per gait cycle, based on previous study Ma et al. (2024), dataset analysis Zheng et al. (2022); Zhu et al. (2021), and the ablation studies in Table 3. Although gait cycles vary across individuals and walking speeds, $L = 16$ serves as a reasonable average Ma et al. (2024). Further experiments on CCGR-MINI shown in Table 2, which covers diverse walking speeds, reaffirm the model's effectiveness under varying temporal conditions. We will explore dynamic estimation of gait cycles for improved adaptability.

### A.4.2 DISCUSSION ON SNIPPET MODELING

In this section, we further compare GaitSnippet with the set-based and sequence-based modeling. Additionally, we analyze the role of Temporal Max Pooling in our framework and discuss potential directions for improvement.

(1) *Comparisons to Set-based Modeling*. The potential to exploit short-term context is a fundamental advantage of the snippet-based paradigm compared to set-based methods. Our framework focuses on leveraging short-term context modeling to enhance frame-level feature extraction and differs from set-based methods Chao et al. (2019); Fan et al. (2023c) in four distinct ways: i) *Sampling*: Snippet sampling *relies on frame order* for sequence partition and *constructs two hierarchical sets* for sampled frames. The outputs are ***not*** permutation-invariant to frame order as clarified above. ii) *Modeling*: We propose an efficient and effective Snippet Block for local context modeling, integrated between two spatial convolutions in a residual block to assist in frame-level feature extraction. iii) *Supervision*: Fine-grained snippet-level supervision is introduced to further enhance the training process. iv) *Performance*: With a backbone composed of 2D convolutions, GaitSnippet significantly outperforms the best set-based method, DeepGaitV2-2D, by a large margin (*e.g.*, R1-+9.3% and mAP-+9.0% on Gait3D).

(2) *Comparisons to Sequence-based Modeling*. The potential to capture long-term dependencies is another fundamental advantage of the snippet-based paradigm compared to sequence-based meth-

ods, as *snippets sampled from a sequence are not temporally continuous and likely cover long-term frames*. As a pioneering attempt, GaitSnippet improves long-term modeling through two key aspects: i) *Fine-grained Snippet-level Supervision*: This encourages the features of snippets from the same sequence to remain similar, even when there is a large temporal interval between snippets. ii) *Diverse Sequence-level Representations*: Unlike sequence-based methods, the input for Temporal Max Pooling at the end of the backbone to derive sequence-level representations changes from continuous frames to snippets that span long-term frames. It significantly enhances the diversity of sequence-level representations for training, which helps the recognition head (*i.e.*, HPM and BN-Neck) better adapt to long-term modeling.

(3) *Role of Temporal Max Pooling*: i) Temporal Max Pooling is indeed an effective manner for feature aggregation in unordered sets Zaheer et al. (2017); Chao et al. (2019); Fan et al. (2023c). Taking GaitSet Chao et al. (2019) as an example, it applies it to all frames per sequence, treating them as a single set. In contrast, GaitSnippet organizes data **hierarchically**, *i.e.*, frames per snippet and snippets per sequence, enabling more structured modeling. ii) GaitSet adopts Temporal Max Pooling *after the backbone* for global aggregation, whereas GaitSnippet integrates it *within the backbone* to enhance frame-level feature extraction via snippet-level context. iii) In our design, Temporal Max Pooling serves as a pioneering component for intra-snippet gathering, consistently yielding performance gains across benchmarks. We will explore more advanced hierarchical temporal aggregation strategies under the snippet paradigm.

(4) *Limitations and Further Improvement*. It is important to acknowledge that our solution for snippet-based gait recognition is not necessarily optimal, and we recognize some limitations in GaitSnippet. For example, in the inference phase, while we achieve superior performance with a single forward process (*i.e.*, $L_1 = 16$), multiple forward processes are required to benefit from different partition strategies (*i.e.*, $L_1 \in \{8, 16\}$). Incorporating partition ensemble into a single forward process represents a meaningful direction for future work. Despite this, our solution consistently shows performance gains across various benchmarks, suggesting that snippet-based gait recognition is a highly promising approach.

## A.5 MORE LITERATURE REVIEW

In Section 2, we primarily review silhouette-based gait recognition methods, which fall under the appearance-based category. For completeness, we also provide a brief summary of model-based gait recognition here.

Model-based approaches primarily focus on explicitly modeling the walking process. While early research in this category relied on hand-crafted features Lee & Grimson (2002); Zhang et al. (2007), recent studies predominantly leverage 2D/3D pose representations Liao et al. (2020); Teepe et al. (2021; 2022); Zhang et al. (2023); Pinyoanuntapong et al. (2023); Guo & Ji (2023); Fan et al. (2024) or SMPL parameters Li et al. (2020; 2022) as input for data-driven feature learning using deep neural networks. For instance, Teepe *et al* Teepe et al. (2021) model 2D poses as graphs and process pose sequences using a Graph Convolutional Network. Fu *et al* Fu et al. (2023) enhance the generalization capability of 2D pose-based gait recognition by applying normalization techniques to the input and extracting fine-grained features. Guo *et al* Guo & Ji (2023) develop a physics-augmented autoencoder framework for 3D pose-based gait recognition. SkeletonGait Fan et al. (2024) converts 2D pose data into a heatmap-like representation, enabling the use of Convolutional Neural Networks for feature extraction.

Moreover, emerging research has focused on fusing multiple modalities Hsu et al. (2022); Peng et al. (2023); Cui & Kang (2023) for gait recognition or exploring new modalities, including RGB images Liang et al. (2022); Ye et al. (2024), point clouds Shen et al. (2023); Han et al. (2022), and event cameras Wang et al. (2019).

## A.6 THE USE OF LARGE LANGUAGE MODELS (LLMS)

In preparing this manuscript, Large Language Models (LLMs) were used solely for checking potential grammatical and stylistic issues in the writing. The use of LLMs did not influence the development of research ideas, experimental design, data analysis, or the interpretation of results. All

scientific contributions, including the methodology, experiments, and conclusions, are entirely the work of the authors.