# OpenReview forum: "GaitSnippet: Gait Recognition Beyond Unordered Sets and Ordered Sequences"
_ICLR.cc/2026/Conference — ICLR 2026 Poster_

### Official Review · Reviewer_yKZm · 2025-10-16

**Soundness:** 3
**Presentation:** 3
**Contribution:** 3
**Rating:** 6
**Confidence:** 5

**Summary:**

This paper proposes a snippet-based gait recognition method aiming to resolve the drawbacks of current mainstream methods, i.e., set-based and sequence-based methods. Specifically, this paper proposes to represent a gait sequence as a combination of individualized snippets which contain several randomly selected frames from a continuous segment of the sequence. The randomly selected neighboring frames make the utilization of local context information better than that of set-based methods, while the randomly selection itself relax the constraint on continuity in sequence-based methods which is difficult to be satisfied in practical situation. Furthermore, this paper proposes rational approaches to snippet sampling and snippet modeling. Through extensive experiments, the effectiveness of this method is demonstrated.

**Strengths:**

1.	The paper is well-written and easy to follow. Most relevant details are described with enough depth providing the reader with enough information to properly grasp the overall idea and the rationale behind the choice of the implemented components. The implementation details seem to be enough to replicate the proposed solution.
2.	The approach seems quite simple, yet it shows interesting performance.
3.	The proposed approach seems can be applied on top of many other gait recognition methods.

**Weaknesses:**

See the Questions below.

**Questions:**

1. From my perspective, the main distinction between GaitSnippet and other multiscale temporal modeling methods lies in its integration of set-based and sequence-based approaches. Although the experimental results demonstrate its effectiveness, my concern is why such a combination is necessary, given that many prior studies have already shown the superiority of sequence-based methods over set-based ones.

2. What would the performance be if a sequence-based modeling approach were also applied within each snippet?

3. The authors propose to model a gait sequence as a composition of individual actions, each represented by an equal-length segment. Would it be more reasonable to represent each action with segments of variable length instead?

4. Sampling with replacement is applied to both segment and frame selection. Could this strategy potentially lead to a loss of temporal information?

---

> ### Author Response · Authors · 2025-11-20
> **Responses to Reviewer yKZm**
>
> Q1: Integration of set-based and sequence-based approaches.
>
> We agree that many prior works have shown the accuracy advantage of sequence-based methods over set-based ones. We propose gait snippets because:
>
> (1) **Limits of sequence-based modeling.** They usually train on short contiguous clips (memory limits) and assume clips are clean and continuous, which often fails in the wild (tracking failures, dropped frames, occlusions). As a result, long-range statistics over the entire track are not fully captured, and the continuity assumption can be brittle in realistic conditions.
>
> (2) **Limits of set-based modeling.** Set-based methods are efficient and robust to missing frames, but treat all frames as an unordered set and cannot explicitly encode short-range temporal cues.
>
> (3) **What snippets provide.** Our snippet formulation combines both: (i) **Sequence-like**: frames in a snippet come from a short contiguous segment, retaining local neighborhoods and short-range context; (ii) **Set-like**: multiple snippets are sampled across the whole track, relaxing strict continuity and enabling modeling of long sequences. With a 2D convolutional backbone, snippet-based sampling and modeling consistently outperform both pure set-based and sequence-based approaches on multiple benchmarks.
>
> ---
>
> Q2: Sequence-based modeling within each snippet.
>
> Our intra-snippet modeling is intentionally set-based: within each snippet, we first apply Temporal Max Pooling (“Gathering”) to aggregate all frames, then a 1×1 convolution (“Smoothing”) with residual fusion back to frame features. This gives each frame short-range temporal context while keeping snippet representations **order-agnostic**, **robust to imperfect sampling**, and **lightweight**.
>
> To address the reviewer’s concern, we replaced this design with sequence-style convolutions similar to DeepGaitV2-P3D by inserting a 3×1×1 temporal convolution inside each snippet. On Gait3D, rank-1 drops from 77.5% to 76.1% despite more parameters and computation. Nevertheless, the resulting model still outperforms DeepGaitV2-P3D. We attribute this to (i) the **snippet paradigm**, which still enables multi-scale temporal modeling within and across snippets; and (ii) **snippet-level supervision**, which remains effective and regularizes snippet representations.
>
> We respectfully clarify that we regard our key contribution as the snippet paradigm for gait, with GaitSnippet as a first instantiation using a 2D backbone and lightweight intra-snippet modeling. Stronger modules are fully compatible and constitute promising future work.
>
> ---
>
> Q3: Equal-length vs. variable-length segments for individualized actions.
>
> We decompose a sequence into equal-length segments. While variable-length segments could be more flexible, our fixed-length design balances **simplicity**, **stability**, and **gait-specific priors**:
>
> (1) **Aligning segment length with gait cycles.** We set L to roughly match the average frames per gait cycle so that (i) each segment tends to cover one full cycle or a major sub-phase, and (ii) snippets from these segments correspond to semantically coherent actions. Ablations on L show that values near the average gait period work best, supporting this simple cycle-aware design.
>
> (2) **Training stability and simplicity.** Equal-length segments yield a regular structure (same number of snippets per sequence at a given length), simplify batching and implementation, and make sampling strategies comparable under the same frame budget. Variable-length segments would require additional heuristics or algorithms (e.g., cycle detection or dynamic segmentation), complicating the pipeline.
>
> (3) **Potential of variable-length segments.** Representing different actions with variable-length segments is appealing, but here we adopt a fixed-length, cycle-aware approximation that already works well. We will explicitly discuss variable-length segmentation as future work.
>
> ---
>
> Q4: Sampling with replacement.
>
> We agree that aggressive sampling with replacement could reduce temporal diversity. In our method it is **used only in restricted training cases** and **never at test time**:
>
> (1) **Only when candidates are few.** Sampling with replacement is enabled only when the number of available segments is smaller than the desired number of snippets, or when frames in a segment are fewer than the desired frames per snippet. Some repetition is then unavoidable, but the impact on temporal coverage is small because the candidate pool is already limited and each snippet still comes from a contiguous segment, preserving local structure.
>
> (2) **Training-only.** Stochastic sampling (with or without replacement) is used exclusively during training to generate diverse snippet combinations. At inference, the full sequence is deterministically partitioned into segments and all frames are used in the snippet hierarchy, so no temporal information is discarded in the final recognition.

---

> > ### Author Response · Authors · 2025-11-26
> > **Follow-up on Rebuttal Feedback for Submission #19087**
> >
> > Dear Reviewer yKZm,
> >
> > I hope this message finds you well. We would like to sincerely thank you for your constructive feedback and for recognizing the contribution our work makes. We also greatly appreciate your positive rating.
> >
> > As the rebuttal phase progresses, we kindly follow up on our responses and would greatly appreciate your feedback on the clarifications provided. If you have any further questions, we would be happy to address them promptly.
> >
> > Thank you again for your time and thoughtful consideration.
> >
> > Best regards,
> >
> > The Authors

---

### Official Review · Reviewer_P9Zy · 2025-10-27

**Soundness:** 3
**Presentation:** 3
**Contribution:** 3
**Rating:** 6
**Confidence:** 4

**Summary:**

This paper focuses on the gait recogntion. Authors introduced a new representation of the gait, snipplet, instead of using the current set or sequence representation. Authors argue that with the new representation, it can both capture both local difference in temporal representation as well as the global shape and pose related difference. Authors also introduces specific snipplet modeling for it and build the gait snipplet model. Authors have compared this method with various of different baselines and show promising performance.

**Strengths:**

+ The paper is well motivated and easy to follow. The introduced gait snipplet is able to combine both the sequential information as well as the set-level unordered information for understanding.

+ Authors have provided pretty useful information in Table 3 for different sampling policy for both set, sequence and snipplet. It is interesting to see that the snipplet is able to yield the best performance across three modalities.

+ Authors also compared the gait snipplet on multiple dataset, which they show best results compared with the other methods listed in the paper.

**Weaknesses:**

- Some of the latest results are not included in the paper, e.g., in [1], authors showed a 77.6/70.3 for R1 and mAP on Gait3D, while gait snipplet shows 77.5 and 69.4. Numbers on GREW is also 85.8/92.6 for R1 and R5 on [1] while it is 81.7/90.9. [1] is introducing a skeleton map for gait recognition, which is also a similar new representation that can be compared with snipplet, and frankly speaking, I think the snipplet idea can also be used on the skeleton map. It would be interesting for authors to apply the idea on [1] both for comparison with the latest state-of-the-art method as well as showing the generalization ability for the introduced method.

[1] Skeletongait: Gait recognition using skeleton maps

**Questions:**

In addition to comparison with some other representations that yields the best performance, it would be great if authors are able to provide time cost and resource analysis that is required for training for at least one of the dataset, such as GREW.

---

> ### Author Response · Authors · 2025-11-20
> **Responses to Reviewer P9Zy**
>
> Q1: Comparison to SkeletonGait [Ref1] and latest results.
>
> We thank the reviewer for pointing out SkeletonGait and SkeletonGait++ [Ref1]. We agree that they represent strong recent baselines, and we are happy to clarify how our setting differs and how our snippet paradigm relates to them.
>
> (1) **Different modality and setting.**
>  SkeletonGait++ is a **multi-modal framework** that fuses silhouettes and skeleton maps, whereas our submission focuses on a **silhouette-only** paradigm. In particular, the silhouette branch of SkeletonGait++ is essentially DeepGaitV2-P3D. Under the *same silhouette-only setting*, our GaitSnippet model **significantly outperforms DeepGaitV2-P3D** while using **fewer parameters and FLOPs**, demonstrating that the snippet paradigm provides a more effective and efficient way to exploit temporal information than the P3D-style sequence modeling used in [Ref1]. We will make this relationship explicit in the revised version so that readers clearly see that GaitSnippet is a stronger and lighter replacement for the silhouette branch used in SkeletonGait++.
>
> (2) **Snippet paradigm is orthogonal and transferable to skeleton maps.**
>  We fully agree with the reviewer that our snippet idea can also be applied to skeleton maps. Building on the original submission, we have conducted additional experiments where we directly transfer the snippet modeling to the skeleton-map setting:
>
> - **GaitSnippet (SkeletonMap)**: a model where we simply replace silhouette inputs with skeleton maps, keeping the snippet sampling and Snippet Blocks unchanged.
> - **GaitSnippet (Silhouettes+SkeletonMap)**:  a **multi-modal extension** that follows the fusion strategy of SkeletonGait++, but replaces its silhouette and skeleton branches with our snippet-based counterparts.
>
> On Gait3D, the results are as follows:
>
> | Method                                    | R1   | mAP  |
> | ----------------------------------------- | ---- | ---- |
> | SkeletonGait                              | 38.1 | 28.9 |
> | **GaitSnippet (SkeletonMap)**             | 40.3 | 31.0 |
> | SkeletonGait++                            | 77.6 | 70.3 |
> | **GaitSnippet (Silhouettes+SkeletonMap)** | 78.8 | 73.6 |
>
> In these experiments, **both variants consistently outperform SkeletonGait and SkeletonGait++**, respectively. This empirically confirms that (i) the snippet paradigm is **not tied to silhouettes**, but is a general temporal modeling scheme that can be plugged into skeleton-map based architectures; and (ii) even in the multi-modal regime, snippet-based modeling brings further gains.
>
> [Ref1] SkeletonGait: Gait recognition using skeleton maps. AAAI 2024.
>
> ---
>
> Q2: Computation cost.
>
> We appreciate the reviewer’s suggestion to provide more details about the computational cost.
>
> (1) **Parameters and FLOPs.** In Appendix A.3.1 we report a detailed comparison of the **parameter count and FLOPs** of different models. Concretely, GaitSnippet introduces only a **slight increase** in parameters and FLOPs compared to DeepGaitV2-2D, while remaining **lighter** than DeepGaitV2-P3D and DeepGaitV2-3D. At the same time, GaitSnippet **consistently outperforms all these baselines** in terms of recognition accuracy on all considered benchmarks. This shows that the snippet paradigm offers a clearly better accuracy–efficiency trade-off than both pure 2D set-based models and heavier 3D/P3D sequence-based models.
>
> (2) **Training time and resource usage in practice.** As the reviewer noted, training time naturally depends on the hardware configuration (GPU model, memory size, I/O bandwidth, etc.). To make the comparison fair, all reported models are trained under **the same hardware and training setup** (same GPUs, batch sizes, and training schedules). Under this setting, training time of GaitSnippet and GPU memory consumption is **comparable to that of DeepGaitV2-P3D**.

---

> ### Comment · Reviewer_P9Zy · 2025-11-20
>
> Thanks you for your reply and added experiments. The new results look very promising and I have updated the rating accordingly. Please consider include a more complete results for the updated gaitsnippet with skeletonmap in the table for camera ready, as it is indeed pushing the SOTA a large step.

---

> ### Author Response · Authors · 2025-11-20
> **Responses to Reviewer's Follow-up and Updated Rating**
>
> We sincerely thank the reviewer for the encouraging follow-up and for updating the rating. We are glad that the additional experiments help clarify the strength and generality of the snippet paradigm. We will carefully refine and improve the manuscript according to your valuable comments and suggestions.

---

### Official Review · Reviewer_6tkX · 2025-11-01

**Soundness:** 3
**Presentation:** 3
**Contribution:** 3
**Rating:** 4
**Confidence:** 3

**Summary:**

This paper presents GaitSnippet, a framework for silhouette-based gait recognition that bridges the gap between unordered set-based and ordered sequence-based approaches. The provided motivation is that traditional set-based models excel at computational efficiency but neglect short-range temporal cues, while sequence-based models capture temporal continuity at the expense of scalability and long-range dependencies. The authors reconceptualize gait as a collection of “snippets” and design a snippet-based pipeline encompassing Snippet Sampling, Snippet Modeling, and Snippet-Level Supervision for gait recognition. Experiments on four public datasets demonstrate consistent performance improvements, achieving state-of-the-art accuracy while maintaining efficiency comparable to 2D models and outperforming more complex 3D backbones.

**Strengths:**

S1) The paper introduces a seemingly new paradigm that redefines how temporal context is encoded in gait recognition. By treating gait as a composition of snippets rather than full sequences or unordered sets, the method unifies short- and long-range temporal modeling.

S2) The pipeline is coherently structured—sampling, modeling, and supervision are all systematically justified and experimentally validated.

S3) The experiments are extensive, spanning multiple datasets and including ablation studies on all major design elements (sampling hyperparameters, snippet block components, supervision weights). The performance gains over both 2D and 3D baselines are convincing and consistent.

S4) The proposed approach achieves better accuracy than heavier 3D models while maintaining a lower computational cost, making it attractive for deployment in real-world gait recognition systems where efficiency is critical.

**Weaknesses:**

O1) While snippets are conceptually appealing, the paper provides limited theoretical justification for why random snippets generalize better than full sequences or unordered sets. The argument about mimicking human perception (recognition from partial cycles) is plausible but not formalized or empirically isolated. It remains unclear whether the improvement stems from better regularization, temporal diversity, or implicit data augmentation.

O2) Although the results are strong, the paper reads as primarily an engineering contribution. There is little analysis of why snippet-level modeling yields gains—e.g., visualization of learned temporal attention, error distribution over gait conditions, or qualitative comparisons of temporal coherence. Such insights would bolster understanding of the snippet paradigm’s internal dynamics.

O3) The architecture largely extends existing 2D residual frameworks by inserting pooling-based temporal operations. The core novelty thus lies more in data organization than in model architecture.

O4) While technically detailed, the paper is verbose and occasionally repetitive (e.g., reiterating motivations for snippets across sections). Also a reader may struggle to identify the primary technical novelty amidst extensive procedural detail.

O5) All experiments are silhouette-based. The paper does not explore how the snippet paradigm could extend to multimodal inputs (e.g., skeletons, RGB) or varying frame rates, which would be important for demonstrating broader impact. Moreover, robustness to occlusion, viewpoint shifts, or missing frames—scenarios where snippet sampling could be advantageous—are not analyzed.

**Questions:**

Please address Weaknesses O1-O4.

---

> ### Author Response · Authors · 2025-11-20
> **Responses to Reviewer 6tkX**
>
> Q1: Theoretical justification.
>
> Thank you for the insightful comment. While a formal theoretical proof is challenging, our snippet formulation is guided by the following principles:
>
> (1) **Unified temporal formulation.**
>
> Snippet-based modeling generalizes existing paradigms:
>
> - If the whole sequence is one segment and all sampled frames form one snippet, our method reduces to a set-based approach.
> - If that single snippet is a contiguous clip, we recover the sequence-based setting with a fixed-length window.
>
> Thus, set-based and sequence-based methods become special cases in a unified “frames → snippets → sequence” hierarchy.
>
> (2) **Multi-scale temporal modeling.**
>
> Snippets enable multi-scale temporal encoding under computational constraints:
>
> - Set-based methods discard local temporal structure and retain only global statistics.
> - Sequence-based methods typically rely on a single fixed-length contiguous clip.
>
> In our design, frame-level features preserve fine-grained spatial details. Within each snippet, residual snippet blocks model **short-range dynamics**; across snippets from all segments, the network aggregates **long-range information**.
>
> (3) **Connection to human perception.**
>
> Prior studies show that humans can recognize individuals from partial motion cues [Ref1]. Our snippet training mirrors this “partial-cycle” setting: the network maps randomly sampled motion fragments to consistent identities.
>
> (4)  **Empirical support.**
>
> Experimentally, snippet modeling with a 2D backbone consistently outperforms set-based and sequence-based variants, while being lighter than 3D/P3D models (Appendix A.3.1).
>
> [Ref1] Neural mechanisms for the recognition of biological movements. Nature Reviews Neuroscience 2003.
>
> ---
>
> Q2: Why snippet modeling yields gains.
>
> In our view, the gains mainly come from:
>
> (1) **Multi-scale temporal modeling.**
> As noted in Q1, snippets allow the model to capture temporal information at multiple scales, which is especially helpful under complex walking conditions (e.g., CCPG and CCGR-MINI).
>
> (2) **Robustness to missing frames.**
>
> We partition each trajectory into contiguous segments and randomly sample frames within each segment to form snippets. This reduces reliance on “perfect” gait cycles and emphasizes recurring cues that survive frame dropping.
>
> To further verify this, on Gait3D, we simulate different frame-dropping levels at test time and report Rank-1 accuracy (%):
>
> | Reserved Frame Ratio | 1/1  | 1/2  | 1/3  | 1/4  |
> | -------------------- | ---- | ---- | ---- | ---- |
> | DeepGaitV2-P3D       | 74.4 | 69.6 | 63.5 | 57.1 |
> | GaitSnippet          | 77.5 | 73.7 | 70.3 | 66.2 |
> | Improvement          | +3.1 | +4.1 | +6.8 | +9.1 |
>
> GaitSnippet consistently outperforms DeepGaitV2-P3D, and the margin widens as fewer frames are reserved.
>
> ---
>
>
> Q3: Architecture design.
>
> We agree that a central contribution is to **rethink the temporal organization of gait sequences**, and our architecture is tightly coupled to this idea:
>
> (1) **Snippet Block as a temporal operator.**
> Residual Snippet Block is more than pooling. It (i) performs non-parametric temporal pooling within snippets, (ii) smooths the snippet representation, and (iii) feeds it back into frame-level features. This converts a plain 2D backbone into a snippet-aware temporal network, where spatial layers operate on features enriched by local temporal context.
>
> (2) **Hierarchical pooling and supervision.**
>  The model is explicitly hierarchical: frame → snippet → sequence. We apply both intra- and cross-snippet pooling and introduce a dedicated snippet-level supervision branch.
>
> (3) **Data–model co-design.**
>  Our contribution lies in jointly designing (i) snippet-based temporal structuring of the data and (ii) a lightweight snippet-aware architecture on top of a 2D backbone. This co-design allows us to outperform 3D/P3D models while remaining efficient.
>
> ---
>
>
> Q4: Writing suggestion.
>
> Thank you for kind suggestion. We will refine carefully.
>
> ---
>
> Q5: More Experiments.
>
> (1) **Beyond silhouettes.**
>
> We apply GaitSnippet to SkeletonMap and to multimodal Silhouettes+SkeletonMap inputs on Gait3D, following [Ref2], and observe consistent gains:
>
> | Method                                   | R1   | mAP  |
> | ---------------------------------------- | ---- | ---- |
> | SkeletonGait                             | 38.1 | 28.9 |
> | **GaitSnippet (SkeletonMap)**            | 40.3 | 31.0 |
> | SkeletonGait++                           | 77.6 | 70.3 |
> | **GaitSnippet (Silhouette+SkeletonMap)** | 78.8 | 73.6 |
>
> (2) **Challenging conditions.**
> The datasets we evaluate on, such as Gait3D and CCGR-MINI, contain occlusions, viewpoint changes, missing frames and varying frame rates. **As noted in Q2**, our controlled Gait3D study with simulated frame dropping shows that GaitSnippet keeps a clear margin over the baseline and degrades more gracefully as the dropping ratio increases.
>
> [Ref2] SkeletonGait: Gait recognition using skeleton maps. AAAI 2024.

---

> ### Author Response · Authors · 2025-11-26
> **Follow-up on Rebuttal Feedback for Submission #19087**
>
> Dear Reviewer 6tkX,
>
> I hope this message finds you well. We would like to express our sincere gratitude for acknowledging the contribution our work makes to the field. We truly appreciate your constructive feedback.
>
> As the rebuttal phase progresses, we kindly follow up on our responses and would greatly appreciate your feedback on the clarifications provided. If you have any further questions, we are happy to address them promptly.
>
> Thank you again for your time and consideration.
>
> Best regards,
>
> The Authors

---

### Official Review · Reviewer_fFgr · 2025-11-05

**Soundness:** 4
**Presentation:** 4
**Contribution:** 3
**Rating:** 6
**Confidence:** 3

**Summary:**

This paper presents a novel snippet-based gait recognition method designed to overcome the limitations of traditional approaches: specifically, the lack of long-range modeling in sequence-based methods and the poor short-range temporal context in set-based methods. The core innovation lies in the sampling strategy, which divides the full gait sequence into $M$ equal-length sub-sequences (snippets), from which $N$ frames are uniformly sampled per snippet. For modeling this input, the method introduces intra-snippet and cross-snippet modules, which primarily adopt the structure of set-based methods, allowing for a fully 2D network architecture. Although the concept of snippet-based modeling is inspired by techniques from action recognition, its application in the gait domain yields a model that is both straightforward and efficient.

**Strengths:**

1. The proposed method achieves improvements in both retrieval performance and inference speed.
2. This paper introduces a simple approach to periodic modeling in gait recognition, which only requires an average estimation of frame length to guide snippet sampling. This is practical for gait recognition;
3. Detailed experiments demonstrate the effectiveness of the sampling strategy and modeling design.

**Weaknesses:**

1. Regarding the framework design, the approach appears incremental and lacks clear differentiation from previous methods. The snippet sampling essentially extends uniform sampling, while the modeling constrains the receptive field through various blocks.
2. Unlike TSN, which targets general video understanding, gait data exhibits inherent periodicity. Is there any analysis examining how this characteristic influences the snippet design?
3. Is the 32-frame setting standard in gait recognition tasks, such as among other SOTA methods in Table 1? What is the length of one complete sequence during inference? How is L_1 determined at inference time?

**Questions:**

Please refer to Weaknesses

---

> ### Author Response · Authors · 2025-11-20
> **Responses to Reviewer fFgr**
>
> Q1: Further clarifications about snippet sampling and modeling.
>
> Our snippet-based framework is indeed inspired by snippet ideas in video understanding, but it differs from prior gait methods and TSN-style designs in **sampling**, **modeling**, and **supervision**:
>
> (1) **Beyond uniform set sampling.**  Existing set-based gait methods treat all frames as a single permutation-invariant set and apply one global set-pooling. In contrast, our snippet sampling **uses temporal order to partition the sequence into segments** of length L, then forms a *hierarchical* set structure (frames → snippets → sequence).
>
> (2) **Snippet modeling inside the backbone, not only at the top.**  TSN-style approaches typically aggregate snippet features only at the final stage. Our **Residual Snippet Block** is inserted *inside each residual block* of the 2D backbone: (i) Intra-Snippet Gathering aggregates features within each snippet; (ii) The snippet representation is smoothed and then **fused back** into each frame through a residual connection.  This means every spatial layer operates on features that already encode local temporal context.
>
> (3) **Snippet-level vs. sequence-level supervision.**  We further introduce **snippet-level supervision** as an auxiliary branch aligned with the snippet hierarchy. This encourages consistency across snippets of the same identity, while incurring no additional cost at inference. Prior set-based and TSN-style gait methods do not exploit such multi-granularity supervision.
>
> Empirically, with *a 2D convolution-based backbone*, our method achieves significant gains over strong set-based and sequence-based baselines across multiple benchmarks.
>
> ---
>
> Q2: The analysis examining the influence of gait periodicity on snippet design.
>
> Gait periodicity explicitly guides how we design the snippet structure, and we will make this connection much clearer in the revised version.
>
> (1) **Snippet length is tied to an approximate gait cycle.**  We deliberately set the segment length L to roughly match the average number of frames in a gait cycle (e.g., L=16). Under this setting, each segment tends to cover one full walking cycle or a large fraction of it, so a snippet sampled from a segment encodes a coherent phase of the gait motion rather than an arbitrary fragment. This design is specific to gait and directly exploits its periodic nature.
>
> (2) **Ablation over L reflects the role of periodicity.**  Our ablations over different L values show that the best results are obtained when L is close to the empirical gait-period length. When L is much smaller or much larger than this range, accuracy degrades. This behavior is consistent with the intuition that L should align with one cycle to capture discriminative periodic patterns.
>
> (3) **Robustness under varying walking speeds and future analysis.**  Although the exact period varies with walking speed and subject, our results on datasets with diverse conditions (including cross-covariate and cloth-changing scenarios) indicate that using a *dataset-level* average period for L is already effective in practice. In the revised paper, we will (i) explicitly discuss how the chosen L relates to dataset gait-cycle statistics, and (ii) clarify that exploring dynamic, per-sequence period estimation is an interesting direction for future work.
>
> ---
>
> Q3: Experimental setting details.
>
> (1) **Is 32 frames a standard setting?**  In training, we use a frame budget of S=32 sampled frames per sequence (e.g., M=4 snippets × N=8 frames each). This budget is in line with many set-based and sequence-based gait methods that operate on around 30–32 frames during training. For fair comparison in our ablations, representative baselines are also evaluated under the same S=32 budget (Table 9 in the Appendix).
>
> (2) **Sequence length during inference.**  At inference time, we **do not subsample** the sequences: all available silhouettes in each sequence are used. The full sequence is partitioned into K segments of length L, and each segment forms one snippet. If the sequence length is not perfectly divisible by L, the remaining frames are treated as an additional segment. Thus, the effective sequence length during inference is simply the full track provided by the dataset (typically ranging from tens to hundreds of frames).
>
> (3) **How is L1 determined at inference?**  During training, we randomly choose the first segment length L1∈{1,…,L} to increase sampling diversity. During inference, we **fix L1=L** so that all segments  (except possibly the last one if there is a remainder) have the same length L. Equivalently, inference corresponds to M=K, N=L with no randomness. We will highlight this setting more clearly in the revised Sec. 3.1.2.

---

> ### Author Response · Authors · 2025-11-26
> **Follow-up on Rebuttal Feedback for Submission #19087**
>
> Dear Reviewer fFgr,
>
> I hope this message finds you well. First, we would like to sincerely thank you for your positive feedback on our submission #19087. We truly appreciate your time and thoughtful comments. As the rebuttal phase progresses, we kindly follow up on our responses and would greatly appreciate your feedback on the clarifications provided. Should you have any further questions, we would be happy to address them promptly.
>
> Thank you again for your time and consideration.
>
> Best regards,
>
> The Authors

---

### Author Response · Authors · 2025-11-30
**Author Responses to AC**

Dear AC,

We sincerely thank all reviewers for their thoughtful evaluations, constructive suggestions, and positive recognition of our work. In the first-round reviews, **three reviewers gave a positive score of 6**, and the remaining reviewer—despite assigning a score of 4—**clearly acknowledged the value of our contribution and raised constructive questions**, all of which we have thoroughly addressed in our Author Responses. Across reviewers, the clarity of the paper, the coherence of the snippet paradigm, the breadth of experiments, and the strong accuracy–efficiency trade-off were consistently highlighted.

Throughout the rebuttal, we addressed each concern in detail. Notably:

- **Novelty and motivation**: Reviewers appreciated that snippets provide a unified temporal formulation bridging set- and sequence-based modeling. We clarified (6tkX, yKZm) that snippets enable multi-scale temporal modeling and improve robustness to real-world tracking noise, supported by new frame-dropping experiments.
- **Design choices grounded in gait periodicity**: We explained how snippet length is tied to gait-cycle statistics and validated this design choice with dedicated ablation studies (fFgr).
- **Generalization beyond silhouettes**: Following reviewer suggestions (P9Zy), we extended GaitSnippet to SkeletonMap and Silhouette+SkeletonMap inputs, outperforming SkeletonGait and SkeletonGait++. This demonstrates that the snippet paradigm is general, transferable, and not restricted to a single modality.
- **Computational aspects**: We clarified training cost and model size (P9Zy), showing that GaitSnippet remains lighter than P3D/3D approaches while consistently achieving superior accuracy.

Reviewers also provided helpful suggestions regarding writing and clarity, which we will incorporate into the camera-ready version.

Importantly, **during the discussion phase—before reviewers were restricted from further interaction—Reviewer P9Zy raised the score from 6 to 8**, explicitly noting that **“the new results look very promising.”** This reflects strengthened confidence in both the novelty and empirical value of our approach.

We sincerely thank the Area Chair for overseeing this submission and hope that our clarifications and additional results help convey the full value and significance of the work.

Best regards,

The Authors

---

### Meta-Review · Area_Chair_LFp8 · 2026-01-03

**Summary:**

The primary concerns raised by the reviewer who initially gave a below-threshold score centered on limited theoretical justification, incremental novelty, and the need for clearer motivation behind the snippet formulation, particularly why random snippet sampling should outperform standard set- or sequence-based modeling. Additional questions focused on gait periodicity assumptions, design choices such as fixed snippet length and sampling with replacement, and whether the approach meaningfully differs from prior multiscale temporal modeling methods. Through the rebuttal, the authors provided substantive clarifications and new experimental evidence, including ablations tying snippet length to gait-cycle statistics, robustness analyses under frame dropping, extensions beyond silhouettes to skeleton and multimodal inputs, and comparisons demonstrating a stronger accuracy–efficiency trade-off than both 2D and 3D baselines. These responses directly addressed the core technical doubts raised by the rejecting reviewer and reframed the contribution as a unified temporal modeling paradigm rather than a minor architectural tweak. While the work remains more engineering- and empirically driven than theoretically formal, the key objections were sufficiently resolved, and the overall reviewer consensus shifted toward borderline positive support, justifying acceptance.

**Reviewer Concerns:**

**Concerns addressed by the rebuttal:**

The rebuttal directly addressed the key concerns raised by the reviewer who initially assigned a below-threshold score. In particular, the authors clarified the novelty of the snippet paradigm relative to prior set- and sequence-based gait methods, explaining how snippets unify short-range and long-range temporal modeling rather than simply extending uniform sampling. They provided new ablation studies linking snippet length to gait periodicity, which addressed questions about the rationale behind fixed-length segments. Additional experiments demonstrated robustness to missing frames, stronger performance under frame dropping, and generalization beyond silhouettes, including extensions to skeleton-map and multimodal inputs. Reviewers’ concerns about efficiency and fairness of comparisons were also addressed through added analyses of parameters, FLOPs, and training cost. These responses resolved the main technical and empirical doubts raised in the initial reviews and led to improved reviewer confidence during discussion.

**Concerns that remain outstanding:**

Some concerns remain only partially resolved. The theoretical justification for why random snippet sampling yields better generalization remains largely intuitive and empirically motivated, rather than formally derived. While the authors provide strong empirical evidence, deeper theoretical analysis of the underlying learning dynamics is still lacking. In addition, although the snippet paradigm is shown to generalize to skeleton and multimodal settings, the paper does not fully explore variable-length snippets, adaptive period estimation, or broader multimodal scenarios, which limits the scope of the current contribution. These remaining issues are best viewed as limitations in depth and scope rather than fundamental flaws, and they do not outweigh the clarified novelty and strengthened empirical evidence provided in the rebuttal, supporting the decision to accept.

**Reviewer Scores:**

Among the reviewers, only one initially provided a below-threshold rating. After carefully considering the authors’ rebuttal, the Area Chair believes that the concerns raised in that review, which are valid, do not constitute critical reasons to reject the paper. The rebuttal substantially clarified the technical motivations and provided additional empirical evidence addressing those concerns. Had that reviewer been able to fully participate in the discussion, it is likely that their score would have increased to a marginally positive level. The other reviewers already held positive views and did not raise new issues. Overall, the final assessment would therefore be uniformly positive but remain borderline, supporting acceptance while acknowledging remaining limitations.

---

### Decision · Program_Chairs · 2026-01-26

Accept (Poster)